# Comparative effectiveness of surgery in traumatic acute subdural and intracerebral haematoma: study protocol for a prospective observational study within CENTER-TBI and Net-QuRe

Thomas A. Van Essen,[1,2] Victor Volovici,[2,3] Maryse C. Cnossen,[2] Angelos Kolias,[4,5] Iris Ceyisakar,[2] Daan Nieboer,[2] Lianne D. Peppel,[6] Majanka Heijenbrok-Kal,[6] Gerard Ribbers,[6] David Menon,[7] Peter Hutchinson,[4,5] Bart Depreitere,[8] Godard C. W. de Ruiter,[1] Hester F. Lingsma,[2] Ewout W. Steyerberg,[1,2,9] Andrew I Maas,[10] Wilco C. Peul[1]

Preliminary portions of this work were presented in a poster presentation at the 11ᵗʰ World Congress on Brain Injury, The Hague, The Netherlands, 2016.

**Correspondence to**
Dr Thomas A. Van Essen;
t.a.vanessen@erasmusmc.nl

## ABSTRACT

**Introduction** Controversy exists about the optimal treatment for patients with a traumatic acute subdural haematoma (ASDH) and an intracerebral haematoma/contusion (t-ICH). Treatment varies largely between different regions. The effect of this practice variation on patient outcome is unknown. Here, we present the protocol for a prospective multicentre observational study aimed at comparing the effectiveness of different treatment strategies in patients with ASDH and/or t-ICH. Specifically, the aims are to compare (1) an acute surgical approach to an expectant approach and (2) craniotomy to decompressive craniectomy when evacuating the haematoma.

**Methods and analysis** Patients presenting to the emergency room with an ASDH and/or an t-ICH are eligible for inclusion. Standardised prospective data on patient and injury characteristics, treatment and outcome will be collected on 1000 ASDH and 750 t-ICH patients in 60–70 centres within two multicentre prospective observational cohort studies: the Collaborative European NeuroTrauma Effectiveness Research in Traumatic Brain Injury (CENTER-TBI) and Neurotraumatology Quality Registry (Net-QuRe). The interventions of interest are acute surgery, defined as surgery directly after the first CT at presentation versus late or no surgery and craniotomy versus decompressive craniectomy. The primary outcome measure is the Glasgow Outcome Score-Extended at 6 months. Secondary outcome measures include in-hospital mortality, quality of life and neuropsychological tests. In the primary analysis, the effect of treatment preference (eg, proportion of patients in which the intervention under study is preferred) per hospital will be analysed with random effects ordinal regression models, adjusted for casemix and stratified by study. Such a hospital-level approach reduces confounding by the indication. Sensitivity analyses will include propensity score matching, with treatment defined on patient level. This study is designed to determine the best acute management strategy for ASDH and t-ICH by exploiting the existing between-hospital variability in surgical management.

### Strengths and limitations of this study

► This comparative effectiveness study is a multicentre prospective observational cohort study that exploits variation in management strategies for intracranial haematomas to enable comparisons of the effectiveness of interventions.

► To overcome the bias by confounding the main analyses uses an instrumental variable approach; this approach is more robust to address unmeasured confounding than conventional individual patients-level analysis methods.

► Large sample sizes will be recruited: 1000 patients with acute subdural haematoma and 750 patients with intracerebral haematoma/contusion are expected, recruited in approximately 70 centres.

► Simulation studies confirmed the expected samples to be sufficient.

► The main limitation of this study is the absence of a randomised assignment of treatment strategy.

**Ethics and dissemination** Ethics approval was obtained in all participating countries. Results of surgical management of ASDH and t-ICH/contusion will separately be submitted for publication in a peer-reviewed journal.
**Trial registration number** NCT02210221 and NL 5761.

## BACKGROUND

In Europe, over two million patients are admitted to hospital each year for traumatic brain injury (TBI), of whom 82 000 people die.[1] Survivors may have long-term physical, cognitive and mental disorders that often necessitate specialised care or rehabilitation programme. This debilitating morbidity has been estimated to lead to enormous societal costs.[2] An acute intracranial haematoma is

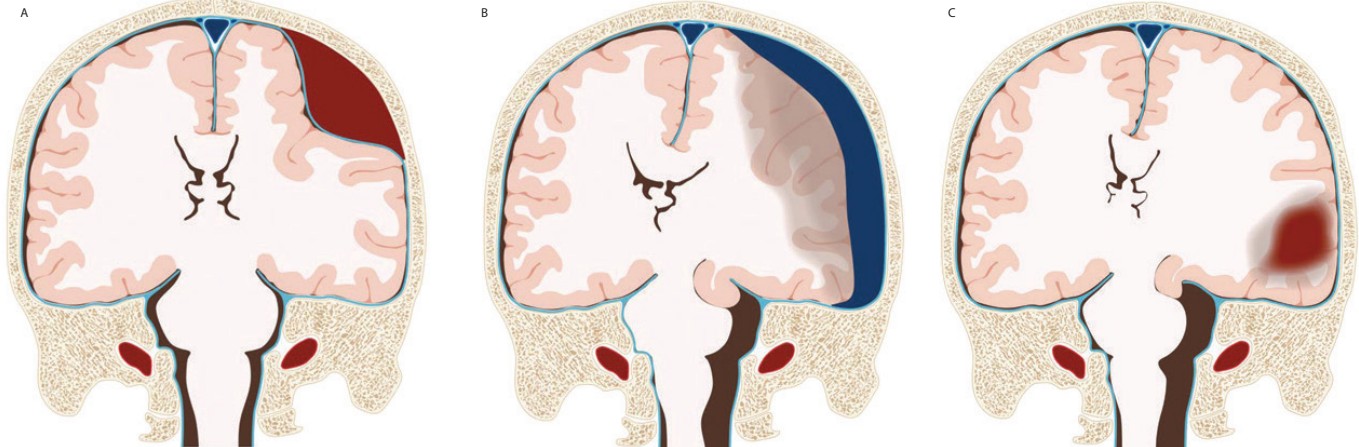

**Figure 1** Different types of post-traumatic intracranial haematoma. (A) Epidural haematoma: a collection of blood between the skull and the outer membrane covering the brain (dura mater). (B) Subdural haematoma: a collection of blood located underneath the dura mater, generally associated with bruising of the underlying brain tissue (contusions). (C) Haemorrhagic contusion and intracerebral haematoma: lesions that reflect similar underlying pathologies, ranging from local bruising (contusions) to bleeding into the brain tissue (haematoma). Figure courtesy of Maartje Kunen, Medical Visuals, Arnhem, Netherlands. Reproduced with permission from Maas *et al*.[48]

the most frequently encountered pathological entity in patients with TBI (figure 1). Predominantly two specific subtypes, the acute subdural and an intracerebral haematoma or contusion (ASDH and t-ICH), occur in, respectively, 11% and 8% of all patients with moderate TBI and up to 49% and 35% of all patients with severe TBI.[3–6] Patients with an ASDH and/or t-ICH can show a wide array of symptoms, ranging from relatively mild complaints such as headache and nausea to severe conditions such as a comatose state (defined by a Glasgow Coma Scale (GCS) <9).

Management of these traumatic haematomas can be challenging and requires the integration of clinical findings and diagnostic imaging. The main question is whether or not the patient needs to be immediately operated on for evacuation of the haematoma and, second, if this surgical evacuation should be accompanied by a decompressive craniectomy (DC, ie, leaving the bone flap out) or not.

Among comatose patients with a large ASDH, direct evacuation of the haematoma leads to a lower mortality.[3 4] Although questioning the effectiveness of surgery in these patients has been compared with questioning the effectiveness of a parachute in skydiving,[5 6] this cannot be generalised to most ASDH patients. Surgery may save a patient's life and preserve neurological function, some, however, may have an unsatisfactory functional outcome, ranging from severe neurological and cognitive deficits to a persistent vegetative state.[7–9] Furthermore, certain subgroups may not benefit from surgery because the damage by the primary injury is simply too devastating.[9] On the other hand, surgery may not always be necessary and a substantial proportion of patients managed conservatively have satisfactory outcomes.[10–14] In addition, timing of surgery plays a role, specifically for a t-ICH. Sometimes a t-ICH is initially managed conservatively,

but may later be treated surgically when a patient deteriorates. The evacuation of the t-ICH can consist of removal of contused brain tissue. Finally, for the decision whether or not the evacuation of the haematoma should be accompanied by a DC, the surgeon weighs the increased complications rate of a DC against the risk of medically intractable diffuse brain swelling.[15]

High-quality evidence for these decisions (if, when and how) is not available. For all guidelines that relate to TBI, the Brain Trauma Foundation (BTF) guideline for surgical management of intracranial haematomas, devised in 2005 by an international panel of experts, is based solely on class III evidence.[4 16] As a result these complex decisions are often based on intuition, regional training and experience of the surgeon, leading to broad practice variation between centres, countries and even between surgeons within a centre.[5 17–21]

Therefore, a systematic evaluation in a (prospective) comparative study is proposed with comprehensive assessment of outcome, including perceived quality of life. We consider an observational comparative effectiveness design the next best alternative to a randomised controlled trial (RCT). These clinical questions are difficult to address in a randomised trial due to several methodological, ethical and pragmatic concerns.[22] Most importantly, the hesitance of clinicians to randomise surgical treatments because of strong opinions on the best treatment hampers realising an RCT.

Here, we present the design of a pragmatic prospective observational cohort study of surgical strategies for ASDH and t-ICH, conducted in the Collaborative European NeuroTrauma Effectiveness Research in TBI (CENTER-TBI) study and the Dutch embedded complete chain of care Neurotraumatology Quality Registry (Net-QuRe).[23 24]

## OBJECTIVES

The primary aim is to compare the effectiveness of acute surgery with expectant management in the treatment of (1) ASDH and (2) t-ICH. The secondary goal is (3) to assess the effectiveness of craniotomy compared with DC for ASDH and/or t-ICH.

## METHODS AND DESIGN

### Design

This study uses a comparative effectiveness research (CER) design, a multicentre prospective observational cohort study that exploits variation in neurotrauma care to create and compare parallel study groups. The multicentre design is necessary to ensure the required number of patients with different neurotrauma treatment strategies for ASDH and t-ICH. The study is conducted in neurosurgical trauma centres in Europe that participate in CENTER-TBI.[23] CENTER-TBI collects data of patients with clinical diagnosis of TBI and an indication for a CT scan.[25] Data for the cohort described in this protocol will partly be collected through CENTER-TBI. The Dutch centres not participating in CENTER-TBI will acquire data through Net-QuRe, in a separate database with a similar data collection protocol.[24 26] The research question and methodology described here were designed before the inclusion of patients.

### Patient and public involvement

Patients were involved in the priority of the research questions and selection of outcome measures in a 'patient advisory panel' consisting of patients and their caregivers. Furthermore, some patients and their caregivers have been asked to join a focus group on the feasibility of the follow-up (burden of the follow-up, the design/length of the questionnaires) and to advise on other research questions. The patient panel and their caregivers will be informed about the developments of the study and will be invited to participate in research meetings and discussions. Also, patients and family are informed of the study results through dedicated websites (Center-tbi.eu and Net-QuRe.nl).

A public debate is going on in the Netherlands about whether or not patients are treated too much at an older age or end-of-life stage. This debate has led to the start of formal campaign 'Choosing Wisely' of which the senior author WPE is organising member.[27] The appropriateness of surgical TBI treatment has been prioritised.

### Eligible study patients

The patients are selected from the observational cohorts from CENTER-TBI or Net-QuRe. Patients presenting to the emergency room with a clinical and radiological diagnosis of ASDH and/or a t-ICH are eligible for inclusion.

The inclusion and exclusion criteria are as following.

### Inclusion criteria

► ASDH and/or a large (>10 cc) t-ICH on a CT-scan.

► Acute presentation (<24 hours of injury) with history of head trauma.
► Clinical indication for admission (ward or intensive care unit, ICU).

### Exclusion criteria

► An ASDH or t-ICH due to penetrating injury, a spontaneous or iatrogenic ASDH/t-ICH.
► Severe pre-existing neurological disorder that would confound outcome assessments.

Patients are not excluded based on other clinical and radiological characteristics (such as: advanced age, antiplatelet/anticoagulant use, small haematoma volume). Radiological criteria for an ASDH or t-ICH/contusion are a high-density lesion, with or without radiological signs of raised intracranial pressure (ICP) and with or without mass effect (ie, midline shift, compression of ventricles and/or basal systems). The minimal size of the t-ICH needs to be 10cc to be included.

### Intervention procedures

Inherent to the observational design of this study the management strategies under investigation proceed according to local emergency and intensive care protocols or surgeon's expertise. Consequently, the resulting variation in management is accepted and analysed. To gain insight into this variation, detailed information is collected on the reasons for specific interventions or management strategies (see section 'why' questions).

### Surgical strategy

Surgical treatment consists of evacuation of the haematoma and/or contusion with a craniotomy, or with a DC, defined as haematoma evacuation plus leaving a large portion of the skull open to allow brain swelling in the secondary phase, preventing subsequent brain injury. Generally, in Europe a craniotomy is performed for haematoma evacuation and DC when (intractable) swelling is seen intraoperatively or when swelling is expected (preventive). The decision for a DC can be made primarily, or secondarily by increasing the defect of the bone flap that is formed during a normal craniotomy. In conjunction to these surgical procedures, the neurosurgeon will decide to place an ICP monitor or not. The ICP device can be an intraparenchymal sensor or an external ventricular drain with a transducer for the ICP. The latter has an option to drain cerebrospinal fluid (CSF) and thereby lower ICP.

The operation will be performed by a qualified neurosurgeon or neurosurgical resident. The techniques for evacuating such haematomas are well established, although specific components of the operations may differ between surgeons. Our aim will be to collect pertinent operative data on a standardised data collection form.

The postoperative care on the ward or ICU generally is protocolised in European centres. The length of hospital stay will differ considerably between patients,

ranging from 1 day to several months, mainly related to the severity of the injury. The aim for a patient is to be discharged as soon as possible. Furthermore, the supportive care is provided as described in the section 'expectant management'.

## Expectant management, with possibly delayed surgery

For patients admitted to the ward, monitoring is in general by clinical neurological control (GCS and motor strength) with or without CT brain follow-up, whereas for those patients admitted to the ICU (mostly severe TBI) the diagnostic and therapeutic options include ICP monitoring with medical management of intracranial hypertension (ie, hyperosmolar therapies, hyperventilation, etc) and cerebral perfusion pressure (CPP).

Follow-up CT scans, performed during hospitalisation, are collected and analysed centrally, independent from the treating physicians. Hereby, an estimation is made about the proportion of the evacuated haematoma and the change in density of lesions.

Patients will be allocated to one of the treatment arms based on the initial treatment strategy. The data collection includes questions after each CT that ask whether or not the patient is transferred to the OR for an operative procedure. In doing so, the treatment 'arms' in this study can be carefully controlled based on the first CT on presentation (showing an ASDH and/or a t-ICH). The initial treatment regimen chosen will be one of either treatment arms, analysed according to an intention-to-treat approach.

## Data collection

Data of care management by hospital personnel are registered in all departments. Data collection is done in a standardised electronic database, based on the 'common data elements' for TBI and web-based data collection protocol.[28 29]

## Practice variation and provider profiling

Parallel to a survey on regional practice variation towards ASDH management in the Netherlands, we have performed provider profiling of neurosurgical care for TBI in Europe.[18] This study provides an exploration of the organisation of neurosurgical care and treatment policy of TBI in all study centres. Such centre characterisation allows specification of the local policy and standardised protocol. Structures and processes of care to be studied are patient volume, location of first Emergency Room (ER) evaluation, level of the trauma centre, referral policy, number of neurosurgeons, type of ICU and 24/7 CT availability. In addition, with regard to postacute care, routine follow-up for ER patients, ICU approaches to fluid load, hyperventilation, hyperosmolar therapy, timing of intracranial surgery, timing of extracranial surgery, glucose management, CSF drainage, DC, CPP management, coagulopathy treatment and ICP monitoring (parenchymal or CSF catheter). Whether or not consensus for divergent clinical decisions is agreed on and whether or not hospital protocols are available and applied.

## Admission data

In short, the 'common data elements' entail the collection of patient characteristics including demographics, comorbidities, associated extracranial injuries, neurological condition, prehospital information, hypoxic and hypotensive periods, and CT abnormalities. For the purpose of CER detailed information on processes of care will be collected within CENTER-TBI and Net-QuRe, including timing (of first CT, second CT, operation), surgery parameters, prehospital management and Intensive Care (IC) therapies (including ICP: pharmacological and/or DC). These data are collected on patient level, but also on hospital level (see paragraph 'Practice variation and provider profiling').

## 'Why' questions

The observational design poses a challenge for inferring that the surgery caused the outcome instead of other factors. Several known and measured confounders (pupillary abnormalities, GCS and haematoma characteristics on CT) can be accounted for. However, for the decision to operate or not in intracranial haematoma, neurosurgeons select certain clinical, radiological and subjective 'gut feeling' characteristics that would normally go unmeasured. Therefore, the following efforts are undertaken to collect variables that normally will go unmeasured. To assess the effectiveness of different treatments for ASDH and t-ICH additional data are collected on an individual (doctor) level; the neurosurgeon, ICU physician and/or neurologist is asked to give their indication/reason to choose for surgery or conservative management (eg, haematoma size, mass effect, clinical symptoms, clinical deterioration and/or other motivation), his or her motivation for the chosen procedure (DC or craniotomy) and the anticipated prognosis of the patient.

The motivation for surgery and the prognosis is collected before the decision for surgery or conservative management because afterwards the motivation could have been changed due to several factors such as intraoperative findings and the clinical course of the patient after the surgery.

Because of the infrastructure in TBI care, in which the clinician on-call is often outside the hospital and—in case of the neurosurgeon—will start with the surgery right after arrival in the hospital, the neurological and neurosurgical residents will assess their supervising clinician's motivation before the decision is carried out. To control this process, the date and time of collecting these variables is collected as well.

## Outcome measurements and endpoints

Within CENTER-TBI and Net-QuRe, the outcome measures are assessed by face-to-face interviews, postal or emailed questionnaires or by telephone interviews at 3, 6

and 12 months after injury. Outcome assessment is done naïve for the research questions.

The primary endpoint is the 6-month Glasgow Outcome Score-Extended (GOSE[30]).

The Glasgow Outcome Scale Extended (GOSE) is the most commonly used outcome measure in TBI. The GOSE grades disability on an 8-point scale incorporating physical deficits as well as emotional and cognitive disturbances affecting disability.[31 32] The GOSE is designed as a structured interview and can be applied through telephone[33] and the mail.[34]

Secondary endpoints are mortality, structural haematoma changes on CT, frequency and type of neurosurgical interventions, ICU and hospital length of stay (days), complications (hydrocephalus, intracranial haemorrhage, infection, pulmonary embolism, deep vein thrombosis and death), 'treatment failure' during the initial hospital admittance (patient in the expectant group who are operated at a delayed moment or patients in the early surgery group who are operated again), discharge to home (from the hospital, rehabilitation facility or nursing home), quality of life 6 and 12 months postinjury with the Short Form (SF)-12, the brain injury specific Quality of Life after Brain Injury Questionnaire[35] and cognitive tests. Details of these and other outcome measures are provided in our previous publication.[23]

## Data analysis

Patient characteristics, hospital characteristics and variation in both treatment and outcome variation will be described using descriptive statistics. To assess differences between groups, appropriate tests will be employed according to distribution and scale of measurement (Student's t-tests or Mann-Whitney U tests for continuous variables and $X^2$ tests or Fisher's exact test for categorical variables). To examine effectiveness of interventions, proportional odds logistic regression models with the 8-point ordinal GOSE as outcome variable will be used. A proportional odds model increases statistical power in comparison to a conventional logistic regression model with a binary outcome.[36] The ORs derived from a proportional odds regression model could be interpreted as the OR for shifting over the GOSE.[36]

The main challenge in the analyses is how to estimate a treatment effect in these observational data with strong confounding. Conventional methods, patient-level analyses with covariate adjustment in regression modelling and propensity score matching, can insufficiently account for the (unmeasured) confounding in TBI.[37] Therefore, the main analyses will use the between-hospital variation in treatment for determining effectiveness by comparing regional treatment strategies. This is an instrumental variable (IV) approach. The instrument is the proportion of patients exposed to the intervention per hospital, determined as the proportion initially operated ASDH of total ASDH patients, initially operated t-ICH of total t-ICH patients and proportion of total ASDH or t-ICH patients exposed to primary DC (that forms a proxy for

'aggressiveness' of the neurosurgical staff). Total ASDH and ICH numbers are available through the registry. The instrument is entered as an independent variable to the analyses. The unmeasured and measured confounding at the hospital level, for example, hospitals that perform more surgery also more often perform other treatments, is overcome with a multilevel analysis model.[38]

In this model, the random intercept should capture the measured and unmeasured confounders at hospital level, resulting in unbiased treatment effect estimates. The random intercept for each hospital represents the unexplained hospital effect (beyond all factors included in the model, including the instrument treatment preference). Assumptions of the IV approach will be checked according to our previous published case study.[37]

For these analyses, hospitals contributing at least 15 patients to the study sample are included to minimise the influence of chance. To increase statistical power, adjustment for potential patient level confounding will be made by adding the strongest predictors of outcome (age, GCS, pupillary response, CT characteristics (haematoma thickness and volume, subarachnoid haemorrhage, basal cistern compromise, other focal or diffuse lesions), hypoxic or hypotensive episodes and extracranial injuries) as covariates.[39] These factors were determined by prognostic modelling in the International Mission for Prognosis and Analysis of Clinical Trials (IMPACT) study based on a dataset of 10 RCTs and three observational studies.[40] Thus, the treatment effect parameter will be the estimate for the effect of 'aggressiveness' on outcome from a random effects ordinal regression model with hospital as a random intercept.

In sensitivity analyses, the instrument validity will be further explored by quantifying a priori collected data, the results of our survey[17] and the provider profiling of CENTER-TBI,[18] and comparing these to the post hoc derived relative proportion exposed to the intervention per hospital.

In the secondary analyses, conventional regression modelling with covariate adjustment and propensity score matching is performed. In both these analyses, actual treatment will be a binary treatment variable and GOSE as ordinal outcome variable. For ASDH surgery effectiveness, confounding will be controlled for by adding age, GCS, pupil reactivity, haematoma thickness and midline shift as covariates in the model. For t-ICH effectiveness, confounding adjustment for age, GCS, haematoma volume, pupil reactivity, haematoma laterality and midline shift. For DC effectiveness, by age, GCS, pupil reactivity, midline shift and haematoma size (ASDH: thickness; t-ICH: volume).

Importantly, the effect of surgery is probably not uniform, as is suggested by empirical evidence[41 42] and by clinical experience. Therefore, effect modification by the following variables will be tested using interaction terms: GCS of ≥9, haematoma size >10 mm in diameter, midline shift >5 mm, time to treatment and baseline prognostic risk.[30]

## Sample size

In CENTER-TBI and Net-QuRe together approximately 1000 patients with ASDH are expected. For t-ICH, 750 patients are expected. Parallel to the core study, 3500 and 3000 ASDH and ICH patients are expected in the registry, respectively. These patients are recruited in approximately 70 centres.

Standard sample size calculations for these specific analyses are not readily available. Therefore, a simulation study was performed to calculate statistical power. The assumptions for these calculations were the following: 30, 50 or 70 hospitals, variation of intracranial operation among hospitals 10%–90% and an effect estimate of OR 0.6 on unfavourable outcome. In addition, we assumed covariate adjustment and the ordinal analysis to increase power with 40%–49%.[36] The simulation confirmed these sample sizes to be sufficient to obtain a power of 80% to detect a difference (assuming a two-sided significance 0.05).

This is in line with preparatory simulation study we performed in which a true treatment effect was simulated specifically to assess the TBI specific associations between covariates and outcome.[37] This simulation study was built around the International and North American Tirilazad trial dataset (86 hospitals between 1992 and 1994) of the IMPACT dataset,[40] which was inflated to CENTER-TBI/Net-QuRe numbers (respectively, 750 and 1000 patients from 70 hospitals). We simulated a hypothetical intervention with an OR 1.5. For the association between the hypothetical intervention and confounders, we used the observed associations between intracranial operation and confounders in the Tirilazad dataset. We used 6-month binary (functional) outcome as dependent variable, which was generated based on a combination of the prognostic effect of the confounders and the effect of the hypothetical intervention.

For analyses and simulations R statistical software with add ons (the rms and lme4 packages) is used.

## Missing data

Missing baseline data will be imputed with multiple imputation (n=5).

## Reporting

Reporting of our study will follow the Strengthening the Reporting of Observational Studies in Epidemiology statement with a special focus on IV analyses recommendations.[43]

## Study limitations

The main limitation of this study is the absence of a randomised assignment of treatment or strategy. Risks of confounding by indication are reduced by IV analysis. The success of the primary analysis, however, depends on the strength of the instrument, that is, the difference in aggressive versus conservative practice style between physicians during the study period. The results of the provider profiling (before the study) are encouraging.

As secondary analyses, we perform more conventional approaches to adjust for confounding by indication (multivariable adjustment and propensity scores). The results of analytical approaches will be interpreted in the light of the assumptions they require and to what extent these are likely to be fulfilled in the data. The final conclusion will be drawn based on the joint results of all analyses.

## DISCUSSION

There is controversy with regard to the initial neurosurgical management of ASDH and t-ICH. First, neurosurgeons are faced with an acute decision to operate or not, and second, are confronted by the choice to evacuate the haematoma with or without a DC. The complexity lies in the balance between too liberal surgical indications with an increased number of survivors with severe disabilities against inappropriate conservative management with unnecessary death and disability. In combination with the circumstances, that is, urgency and time pressure as well as absence of peer consultation, these treatment decisions have been shown to lead to variation in surgical treatment between surgeons.

The proposed study will provide a strong level of evidence for surgical management of ASDH and t-ICH. We expect that the large natural existing practice variation in management of these intracranial lesions[17 18] could in part explain the unexplained between-centre variability in outcome in TBI.[44] Thereby, the impact on patients will probably be significant. Recognising and implementing the most effective clinical treatment strategy could be an important step towards reducing the widely differing injury mortality rates across Europe.[45]

Current and ongoing studies are sparse. Since the BTF guideline, which was based on merely retrospective studies with small or selected study populations that were performed more than 10 years before the guideline,[5 10 11 46] there have been only some comparative studies. In our own retrospective analyses, early ASDH evacuation might be associated with lower odds of mortality and unfavourable outcome (GOS ⩽3).[41] This is the first report showing an effect estimate of surgery for ASDH. The clinical effectiveness of an early evacuation for t-ICH was challenged in the Surgical Trial in Traumatic Intracerebral Haemorrhage Trauma, an international multicentre pragmatic RCT.[47] The study started in October 2009 but was halted due to a disbalance in recruited patients per country. In the analysis of the included patients, a strong (but non-significant) tendency towards benefit of early surgery was found on the primary endpoint the dichotomous GOS and there were significantly more deaths in the initial conservative treatment group. The effectiveness of a primary DC in patients with ASDH is currently being assessed in the recruiting RESCUE-ASDH randomised trial.[15]

Thus, these traumatic haematomas confront the neurosurgeon with a challenging surgical decision-making task,

which, most likely due to a lack of general evidence, leads to broad variation in current surgical practice patterns. While RCTs can be delivered and provide high level evidence, they are challenging to conduct, hence the rationale for other methodological intervention paradigms. The high-quality observational study presented in this article, with a focus on the analysis of the differences in management and outcome, is expected to provide the much-needed further evidence in the field of surgical management of traumatic focal lesions.

### Ethics and dissemination

Written informed consent is obtained from the patient or the legal first representative on the ER and preferably as soon as possible. Center-TBI is registered at ClinicalTrials. gov (NCT02210221) and Net-QuRe is registered at the Netherlands Trial Register (NL5761).[25 26]

Results of surgical management of ASDH and t-ICH will separately be submitted for publication in a peer-reviewed journal. The completion dates of CENTER-TBI and Net-QuRe are 1 April 2020 and 1 July 2022, respectively.

The data that support the findings of this study are available but restrictions apply to the availability of these data, which were used under licence for the current study, and so are not publicly available. Data are, however, available from the authors on reasonable request, with permission of the CENTER-TBI and Net-QuRe management teams.

**Author affiliations**
$^1$University Neurosurgical Centre Holland, Leiden University Medical Centre, Haaglanden Medical Centre and Haga Teaching Hospital, Leiden and The Hague, The Netherlands
$^2$Centre for Medical Decision Sciences, Department of Public Health, ErasmusMC - University Medical Centre Rotterdam, Rotterdam, The Netherlands
$^3$Department of Neurosurgery, ErasmusMC - University Medical Centre Rotterdam, Rotterdam, The Netherlands
$^4$Division of Neurosurgery, Department of Clinical Neurosciences, University of Cambridge and Addenbrooke's Hospital, Cambridge, UK
$^5$NIHR Global Health Research Group on Neurotrauma, University of Cambridge, Cambridge, UK
$^6$Rijndam Rehabilitation and Department of Rehabilitation Medicine, ErasmusMC - University Medical Center Rotterdam, Rotterdam, The Netherlands
$^7$Division of Anaesthesia, Addenbrooke's Hospital, University of Cambridge, Cambridge, UK
$^8$Department of Neurosurgery, University Hospital KU Leuven, Leuven, Belgium
$^9$Department of Biomedical Data Sciences, Leiden University Medical Centre and Haaglanden Medical Centre, Leiden and The Hague, The Netherlands
$^{10}$Department of Neurosurgery, Antwerp University Hospital and University of Antwerp, Edegem, Belgium

**Contributors** TVE conceived of the study design, performed the statistical analysis and wrote the manuscript. GdR, VV, MCC, LP, HL, BD, MH-K, GR, DM, PH, AK, ES, AIM and WP participated in the design of the study and helped to draft the manuscript. DN, IC, HL and TVE performed the simulation studies. All authors read and approved the final manuscript.

**Funding** The project described in this study protocol is funded by the Hersenstichting Nederland (the Dutch Brain Foundation, grant number ps2014.06) for Net-QuRe and the European Union seventh Framework Program (grant 602150) for CENTER-TBI. Additional support for CENTER-TBI was obtained from the Hannelore Kohl Stiftung (Germany), from OneMind (USA), from Integra LifeSciences Corporation (USA) and from NeuroTrauma Sciences (USA). Authors PH and AK are supported by the NIHR Global Health Research Group on Neurotrauma, which was commissioned by the National Institute for Health Research (NIHR) using UK aid from the UK Government (project 16/137/105). PH is supported by a Research Professorship from the NIHR, the NIHR Cambridge Biomedical Research Centre and the Royal College of Surgeons of England. AK is supported by the Royal College of Surgeons of England and a Clinical Lectureship, School of Clinical Medicine, University of Cambridge.

**Disclaimer** The funding bodies had no role in the design of the study, collection, analysis, and interpretation of data and in writing the manuscript. The views expressed in this publication are those of the author(s) and not necessarily those of the NIHR or the Department of Health and Social Care.

**Competing interests** None declared.

**Patient consent for publication** Not required.

**Ethics approval** Ethical approval for the studies CENTER-TBI and Net-QuRe has been received from each participating centre in the study before recruitment started. The list of sites, ethical committees, approval numbers and approval dates can be found on the website: https://www.center-tbi.eu/project/ethical-approval.

**Provenance and peer review** Not commissioned; externally peer reviewed.

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
