## [Reviewer comments · BMJ Open]

ARTICLE DETAILS

TITLE (PROVISIONAL)	Comparative effectiveness of surgery in traumatic acute subdural and intracerebral hematoma: study protocol for a prospective observational study within CENTER-TBI and Net-QuRe
AUTHORS	Van Essen, Thomas; Volovici, Victor; Clossen, Maryse C.; Koliass, Angelos; Ceyisakar, Iris; Nieboer, Daan; Peppel, Lianne; Heijenbrok-Kal, Majanka; Ribbers, Gerard; Menon, David; Hutchinson, Peter; Depreitere, Bart; de Rooter, Godard; Lingsma, Hester; Steyerberg, Ewout; Maas, Andrew I; Peul, Wilco

VERSION 1 – REVIEW

REVIEWER	Stephen Honeybul Sir Charles Gairdner Hospital Australia
REVIEW RETURNED	14-Aug-2019

GENERAL COMMENTS	A well defined important study
--------------------------------

REVIEWER	Janne Koskimäki, MD, PhD The University of Chicago, Neurosurgery, Neurovascular Surgery Program, USA.
REVIEW RETURNED	05-Sep-2019

GENERAL COMMENTS	I have reviewed the manuscript entitled “Comparative effectiveness of surgery in traumatic acute subdural and intracerebral hematoma: study protocol for a prospective observational study within CENTER-TBI and Net-QuRe” by Van Essen et al. Authors describe a protocol for a prospective multicenter observational study aimed at comparing the effectiveness of different treatment strategies in patients with acute subdural hematomas and/or traumatic intracerebral hemorrhages. There are no ethical concerns. Statistical analyses and plan are well explained, and weaknesses of the study design have noted and managed. Sample size seem feasible to obtain in multicenter settings. Study protocol is feasible and scientific background relevant. Aims are clearly presented. Primary and secondary outcome measures are properly defined. Figure is clear. Literature is properly cited. The authors have constructed well described and conceptualized study protocol. Overall, this protocol is well designed and written, and meets high scientific standards. I have only one minor request: Could authors clarify the expected timeline of the study (from start to end until reporting).
--

REVIEWER	John Paul Kolcun, MD Rush University Medical Center, USA
REVIEW RETURNED	15-Sep-2019

GENERAL COMMENTS	The results of this study should inform immediate management decisions for patients with acute hemorrhage following TBI. Secondary analyses of the aSDH and ICH sub-groups will be of particular interest, given the stark differences in presentation, prognosis, and general operative indications between these two entities.
--

REVIEWER	Pepijn van den Munckhof Department of Neurosurgery, Amsterdam University Medical Centres, The Netherlands
REVIEW RETURNED	16-Sep-2019

GENERAL COMMENTS	Highly interesting research protocol that will address very urgent and still unanswered questions concerning the treatment of acute subdural and intracerebral hematoma. Indeed, after all these years of neurosurgical research efforts we still do not know:  - which traumatic intracranial hematomas should be treated - at what stage during the clinical course of these patients the surgery should be performed, and - which operative technique is to be applied The current protocol will address these issues. I am looking forward to the results.
---

VERSION 1 – AUTHOR RESPONSE

Reviewer 1 commented ‘A well defined important study’ and expressed no further comments.

In response to reviewer 2’s request ‘to clarify the timeline of the study (from start to end until reporting)’, the timeline has been clarified in the main text now as well. Reviewer 2, furthermore, indicated that the aims, design, data collection and analyses of the study were well explained and assessed as being feasible.

Reviewer 3 expressed no concerns. We agree with the reviewer’s assessment that the subgroup analyses of ASDH and ICH will be of particular clinical interest.

And finally, we fully agree with reviewer 4 that the study will shed light on whether or not, when and how patients with these intracranial hematomas should be operated. He had no further comments.
